# Sustainability by Function (SbF): A Case Study in a Rainfed Vineyard to Reduce the Loss of Soil Nutrients

Manuel López-Vicente [1,*], Sara Álvarez [2], Elena Calvo-Seas [3] and Artemi Cerdà [4]

1. AQUATERRA Research Group, Advanced Scientific Research Centre, University of A Coruña, Campus de Elviña, 15071 Coruña, Spain
2. Unit of Woody and Horticultural Crops, Instituto Tecnológico Agrario de Castilla y León (ITACyL), Ctra. Burgos km 119, 47071 Valladolid, Spain; alvmarsa@itacyl.es
3. Independent Researcher, Casa Calvo, 22144 Bierge, Spain; eleseas@gmail.com
4. Departament de Geografia, Universitat de València, Blasco Ibàñez, 28, 46010 Valencia, Spain; artemio.cerda@uv.es
* Correspondence: manuel.lopez.vicente@udc.es

**Abstract:** The effectiveness of a seeded cover crop to minimize soil nutrient losses was evaluated in a rainfed vineyard. Two sediment tanks were installed (ST2: drainage area with high ground cover (GC: 82%) and ST3: very high GC (89%)) and samples from 26 time-integrated periods (TIP) were collected over 15 months. The average soil nutrient content was previously estimated in the drainage areas of ST2 ($N_{total}$: 0.967 mg/g; $P_{ava}$: 0.411 mg/g; $K_{ava}$: 1.762 mg/g) and ST3 ($N_{total}$: 0.711 mg/g; $P_{ava}$: 0.437 mg/g; $K_{ava}$: 1.856 mg/g). The sediment nutrient concentrations and the sediment/soil enrichment ratios were comparable between ST2 and ST3, but the total loss of nutrients clearly differed among areas. The loss of nutrients in the area with lower GC (379.7 g N-P-K/ha/yr) was 8.3 times higher than in the area with higher GC (45.8 g N-P-K/ha/yr), and this pattern remained during the months with low, medium and high GC: 91.9, 2.1 and 2.1 g N-P-K/ha/month in ST2 and 6.9, 3.0 and 3.5 g N-P-K/ha/month in ST3. The benefits of greater GC promote the environmental and agronomic sustainability by the functions of the cover crop, favoring healthy soils and a reduction in the investment of the farmers in fertilizers. This is very relevant in a postpandemic world under the threat of the war in Ukraine, the lack of fertilizers and the need for a local production of food.

**Keywords:** vineyard; cover crop; sediment yield; enrichment ratio; sustainability





## 1. Introduction

### 1.1. Soil and Nutrient Loss in Cropland

At the global scale, soil loss (expressed as soil denudation in mm/yr) in arable land ($\bar{x}$ = 0.5 mm/yr) is much higher than that in forests ($\bar{x}$ = 0.2 mm/yr), agroforestry ($\bar{x}$ = 0.1 mm/yr) and other forms of seminatural vegetation ($\bar{x}$ = 0.2 mm/yr) but lower than that of bare soil ($\bar{x}$ = 1.2 mm/yr) [1]. Historically, and in Mediterranean countries, vineyards and other permanent crops such as almond and olive orchards have been cultivated into marginal lands, including steep, stony hillslopes, sometimes on unstable bench terraces, thus leading to increased soil erosion, particularly during intense rainstorms [2]. In a review study performed with data from Spanish and Italian vineyards, the average erosion rate measured by means of different erosion methods was 9.3 Mg/ha yr, which is much higher than the tolerable soil loss rate [3]. Regarding nutrient loss in Mediterranean vineyards, traditional management (weeds removal and ploughing) have favored land degradation and diffuse pollution, albeit (over-)fertilization may have a strong effect on nutrient exports [4]. Additionally, soil and nutrient loss in vineyards affect soil biodiversity, water nutrition and chemical fertility, leading to lower crop yield and quality [5].

One of the most commonly used metrics to assess the actual impact of soil erosion on soil chemical parameters is the sediment/soil enrichment ratio (ER). In the last half-century,

work began to study the ER for organic carbon and matter, clay, nitrogen, phosphorous and other nutrients in different countries, such as in Nigeria [6], USA [7], China [8], Australia [9] and Thailand [10], and nowadays this metric is being used worldwide such as in Indonesia [11], India [12], Spain [13,14] or Brazil [15] for distinct land uses, including annual and permanent crops and crop rotations. In most cases, and in particular in arable land, the ERs were higher than 1.0, highlighting the loss of topsoil fertility. There is a certain selectivity during the erosional and sediment delivery processes involved in the transfer of eroded materials from the hillslopes to the main water bodies. However, some of these authors found that time variation in ER (greater or lower than unity) was largely due to time variation in the sediment size distribution, settling velocity and the predominant mechanism of soil erosion (e.g., detachment by runoff or splash), highlighting the complexity of these processes. However, only a few studies have estimated the sediment soil enrichment ratio in vineyards, such as the study of Farsang et al. [16] in Hungarian vineyards (ER of organic matter, $P_2O_5$, Ni, Cu, Pb, Zn, Co and Cr) and of Ruiz-Colmenero et al. [17] in Spanish vineyards (the ER of organic matter, nitrogen and phosphorus). Therefore, there is a clear necessity for increasing the number of studies that deal with this topic. This study aims to shed light on the process of soil nutrient decline in vineyard soils and to provide new data to reinforce our comprehension of the role of sustainable tillage practices.

### 1.2. Sustainability by Function

In the last decades, specific concepts and ideas have emerged aiming to highlight the key and different roles played by soils. Among them, the concepts of soil security, sustainability, awareness, governance, health and ecosystem services are probably the most useful to promote the best management practices, compromising social and political actors and involving economic and environmental aspects [18]. Soils have a great potential to contribute to sustainable development, and it is recognized that poorly managed, degraded or polluted soils may contribute negatively to both Nature's Contributions to People (NCP) and the United Nations Sustainable Development Goals (SDGs), and, on the contrary, healthy soils are capable of playing vital roles and making positive contributions [19]. Nowadays, we can find an extensive list of measures and practices to preserve soil and improve soil quality, for instance, man-made (e.g., soil bunds, ridges and microtrenches) and natural (e.g., wetlands, floodplains and grassed waterways) sediment sinks reduce outflow of suspended sediments [20]. In farmland with annual crops, conservation tillage (reduced/no-tillage with/out residues) is beneficial to the improvement of integrated soil fertility and crop yield [21]. In permanent crops (e.g., vineyards, olive groves and other fruit-tree orchards), the use of ground covers (e.g., spontaneous vegetation, seeded permanent/temporary cover crop (CC), mulching and agro-textile) has demonstrated its effectiveness to reduce runoff yield, increase surface water infiltration [22] and minimize the loss of soil [23], soil organic matter [13], soil organic carbon [14] and nutrients [24].

The concept of 'Sustainability by Design' (SbD) is mainly used in disciplines related to industry, architecture, energy and business [25,26], but it is not included in environmental studies. The term SbD refers to how sustainability can be achieved through specific designs regarding raw materials supply, money and travel expenditure, life cycle of materials, unintended environmental consequences, waste management and circular economy [27]. In this study, we use the concept of SbD to introduce the concept of 'Sustainability by Function' (SbF) that is defined as the 'enhancement of any indicator that favor the sustainability of any environment and/or socio-economic issue by means of the processes and mechanisms associated with a specific material and/or management'. Aspects such as the geometry and composition of the material and their evolution over time—including lifespan—and system compartments are included in the concept of SbF.

### 1.3. Hypothesis, Objective and Expected Contribution

Based on the existing literature, we expected that within the same vineyard, the zone with a higher ground cover—with a mix between temporary seeded cover crop

and spontaneous vegetation—will have lower sediment/soil nutrient enrichment ratios than the vineyard zone with a lower ground cover where the soil nutrient loss will be more pronounced. To test this hypothesis, we selected a rainfed vineyard in north-eastern Spain where two sediment traps were installed. This study sheds light on this topic, providing new data. In addition, we hope that the use of the concept 'Sustainability by Function' in environmental and agronomic research will boost the use of those materials and management practices that could improve the resilience of farmlands and agro-ecosystems to the circumstances that threaten their sustainability.

## 2. Materials and Methods

### 2.1. Study Area and Tillage Practices

A small vineyard located in north-eastern Spain (42°02′04″ N; 0°04′13″ E), near Barbastro (in an area so-called "Los Oncenos") province of Huesca was selected to perform this study (Figure 1a). The vines were planted in 2008 and have been managed since then by Fábregas winery, a winery with the Certificate of Origin: DOP Somontano. Vines correspond to the Spanish variety Grenache ('Garnacha Tinta'; *Vitis vinifera* L. cv. Grenache). The plantation is arranged in straight lines (espalier system) with an average distance between the grapevine lines (row hereafter) of 3.0 m and of 1.3 m between vines in the same row. Following the traditional practices in this region, viticulturists raised the soil in the rows to be ca. 13 cm—on average—higher than the soil located in the interrow areas. Owing to this practice, overland flow defines a characteristic pattern with straight lines during most rainfall-runoff events, but runoff generated during heavy rainfall events may exceed this topographic threshold. The vineyard is located on a rolling landscape, and elevation ranges from 447 to 468 m a.s.l.

A basic fertilization was conducted just before planting the vines, and no further treatment was applied. No irrigation system was used, and the only source of soil water is rainfall. Therefore, the soil and sediment nutrient content is representative of the natural dynamic of soil redistribution and farmland practices. In this area, the climate is continental Mediterranean. The mean annual precipitation is ca. 621 mm, mainly concentrated in spring (April–June; 31% of the total annual rainfall) and autumn (September–November; 28%), and the mean annual potential evapotranspiration is 1045 mm (data source: 'Oficina del Regante'; Government of Aragon). The summer is dry and hot (mean temperature in July and August of 24.4 °C) with occasional thunderstorms, and snowfall events are scarce in winter (mean temperature in January of 3.8 °C). The mean annual temperature is 14.3 °C.

The interrow areas of the vineyard present a mixture of plant species, including spontaneous vegetation and a plantation of common sainfoin (*Onobrychis viciifolia* Scop., 1772). Spontaneous vegetation also appears in the corridors between the four vineyards of the Los Oncenos subcatchment. The cover crop (CC) was seeded for the first time in early 2016. In spring, and to minimize water and nutrient competition between the plants and the vines, the farmer does one or two mowing pass/es per year, depending on the growth of the ground cover. In this study, the mowing was carried out in the third week of May 2017 (later than usual) and in the first week of May 2018. Most pruning residues remain in the same place after mowing, so the ground cover (GC: percentage of the soil surface covered by living vegetation, leaf litter and pruning residues) remains between moderate and high all over the year. Due to tillage practices (e.g., grape harvest and mowing passes) and the characteristic water scarcity during the summer, important changes in the soil surface cover were observed over the twelve months of the year. Additionally, the phenology of the grapevines and of the seeded cover crop (CC) and spontaneous vegetation influenced GC (Figure 1b). Herbicide was only applied under the vines along the row to avoid the growth of weeds. Grapes were harvested in September. Previous studies in the 'Los Oncenos' subcatchment have demonstrated the effectiveness of the two types of ground cover to favor soil water infiltration, increase the soil water content [28] and reduce the loss of soil and sediment turbidity [29] and of soil organic matter [13].

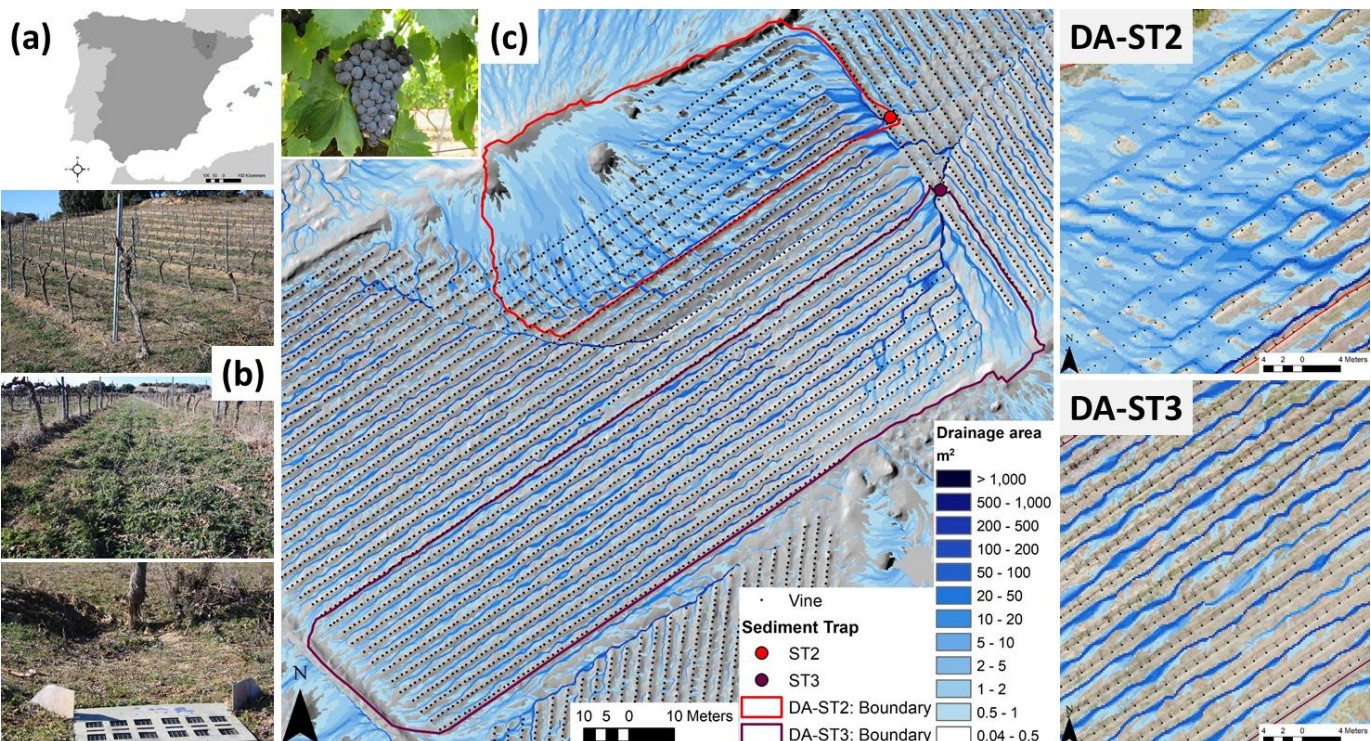

**Figure 1.** Location of the study site in the province of Huesca, NE Spain (**a**). Pictures of the drainage area of ST2 and ST3 and of the sediment trap (**b**); these photos were taken on 12 February 2018 when vegetation cover was limited due to winter conditions and pruning residues still remain on the ground. Map of the overland flow pattern in the study area, showing the boundaries of the drainage areas (DA) of the two sediment traps (ST) (**c**); the maps on the right side show in detail these patterns (more details about the overland flow pattern in [30]).

### 2.2. Sediment Traps and Study Period

To collect runoff and sediment samples in the 'Los Oncenos' subcatchment, two sediment traps (ST) were located in the cereal fields (ST1 and ST4) and two STs in the vineyards (ST2 and ST3). STs were installed in December 2016 and tested in January 2017. ST2 and ST3 were established in the first interrow of the vineyard located near the outlet of the subcatchment, down the corridor that separates the two vineyards (Figure 1c). Each ST was located in the course of an ephemeral gully and buried—installed below the soil surface—in order to avoid any disturbance with the tractor traffic (Figure 1b). The two STs collected the runoff and sediment generated during the different rainfall-runoff events. Each trap was designed to hold a maximum volume of 32.2 L (460 mm length × 200 mm width × 350 mm depth) and had two boxes: one box was buried and remained in the field, and the other one was moveable and located inside the first box. This allowed an easy measurement of the runoff and sediments collected during most of the rainfall events. More details about the pieces and design of the ST can be found in [29].

The drainage area (DA) of ST2 and ST3 covers 3286 m² and 6214 m², respectively, with a mean slope gradient of 17.0% and 9.2%. Despite these differences between the mean slope gradients, the mean slope near the STs was very similar in the two ST-DAs (ca. 9.5%). Additionally, the mean slope of the rows (where bare soil conditions were predominant in the two ST-DAs) was also comparable in ST2-DA and ST3-DA. The highest values of slope gradient in ST2-DA only appeared in the small forest patch ($\overline{S}$ = 18.4%) and in the interrow areas near the forest (Supplementary Figure S1). Therefore, we consider that the impact of the different slope gradients on the total sediment yield and nutrient loss is less relevant than the differences in the percentage of ground cover. The main land use in the two DAs is vineyard (57% and 87% in ST2-DA and ST3-DA), but other land uses occupy

relevant parts within the DAs, such as forest and bare soil trails in ST2-DA and grassed trails in ST3-DA (Table 1). In a previous study, López-Vicente and Álvarez [30] found that intense soil erosion processes have launched the development of continuous flow path lines, breaking the topographic sills of the rows in some sections, in particular in ST2-DA, while parallel and straight flow lines are the main pattern in ST3-DA (Figure 1c: Zoom-in maps). Soils are Haplic Calcaric Regosols in the upper part of DA-ST2—where the small forest patch appears and in the most elevated zone of the vineyard—and Luvic Calcisols (CLl) in the lower part of the drainage area of ST2 and in the whole area of ST3-DA. Regarding the ground cover (by live plants and inert materials), the soil surface of the ST2-DA had less vegetation with an average percentage of bare soil of 18.4%, whereas the ST3-DA only had 10.6% of the soil surface without vegetation.

**Table 1.** Main physiographic and soil characteristics of the drainage area (DA) of the two sediment traps: ST2 and ST3.

| ST | Area | Slope | | | | | | | | | Bare Soil |
|---|---|---|---|---|---|---|---|---|---|---|---|
| | m$^2$ | Mean (%) | Near the ST (%) | IR (%) | R (%) | VY (m$^2$; %) | GT (m$^2$; %) | F (m$^2$; %) | BST (m$^2$; %) | CF (m$^2$; %) | Mean (%) |
| ST2-DA | 3286 | 17.0 | 9.8 | 14.2 | 15.7 | 1889; 57% | 200; 6% | 738; 22% | 404; 12% | 54; 2% | 18.4 |
| ST3-DA | 6214 | 9.2 | 9.2 | 8.4 | 11.7 | 5461; 87% | 581; 9% | 106; 2% | 120; 2% | 0 | 10.6 |

IR: interrow area; R: row; VY: vineyard; GT: grassed trail (corridor between vineyards); BST: trail (bare soil); F: forest; CF: cereal field.

In this study, the observation period lasted 15 months, from February 2017 to April 2018. Regularly, and after each heavy rainfall event or after several low- or medium-intensity rainfall events, the runoff and sediment samples were collected and associated to the corresponding time-integrated period (TIP). The duration of every TIP was measured in the days between the latest and the new field survey. In total, we conducted 26 field surveys. All runoff and sediments were transported to the laboratory, where the total runoff with sediments of each trap was weighted, and sediments were separated by decantation. The samples of wet sediment were dried in an oven at 60 °C for 96 h to ensure a complete dry out of the samples without altering the stability of the chemical compounds. In two recent studies, the values of total runoff (Q; L/TIP), sediment yield (SY; g/TIP) and loss of organic matter (in % and g/ha TIP) were calculated for each ST and TIP [13,29]. In these studies, the number of rainfall events per TIP and their corresponding rainfall depths (R; mm) and erosivity (EI$_{30}$; MJ mm ha$^{-1}$ h$^{-1}$) were calculated. Using this data, we found that the monthly and seasonal changes of rainfall erosivity and ground cover (GC) described the following pattern: low erosivity (median value: 9.0 MJ mm/ha h TIP) during the TIPs with very low and low GC (from June to October; 5 months), moderate erosivity (median value: 11.5 MJ mm/ha h TIP) during the TIPs with medium GC (from November to mid-February; 3.5 months) and higher erosivity (median value: 29.9 MJ mm/ha h TIP) during the TIPs with high and very high GC (from mid-February to May; 3.5 months). These conditions favored soil protection during most erosive events, but some intense events also happened when soil surface was more exposed to soil erosion by water.

*2.3. Soil and Sediment Nutrient Content*

In a previous study and before installing the sediment traps, 222 topsoil samples were collected in 74 sampling points—with three replicates of 250 cm$^3$ per point—covering the different land uses and field compartments of the 'Los Oncenos' subcatchment [28]. Different soil physical (e.g., bulk density, rock and clay content, texture and permeability) and chemical (content of organic matter) parameters were determined at the subcatchment scale (Table 2) [13,29]. In this study, the values of total nitrogen and available phosphorous and potassium are presented for the first time. Regarding the sediments and after drying the samples, the coarse fragments were removed (mean diameter higher than 2 mm) of the fine fractions of the sediment were analyzed in the certified laboratory 'Centro Tecnológico

Agropecuario Cinco Villas S.L.' (Ejea de los Caballeros, Zaragoza, Spain) that is an official center for the analysis of soil, water, organic and inorganic fertilizers, manure, foods, plant tissues and other products related to agronomy. In order to reduce analytical errors, both the soil and sediment samples were analyzed in the same center following the same methods and protocols.

**Table 2.** Mean values of the soil physical and chemical parameters of the drainage area of the two sediment traps: ST2 and ST3.

| ST-DA and Land Use | BD [1] | Rocks [1] | Clay [1] | Texture [1] | SOM [1] | $TN_{soil}$ | $P_{ava-soil}$ | $K_{ava-soil}$ |
|---|---|---|---|---|---|---|---|---|
| | g/cm$^3$ | %$_{weight}$ | % * | Class | % | mg/g$_{soil}$ | mg/g$_{soil}$ | mg/g$_{soil}$ |
| ST2-DA | 1.46 | 19.5% | 7.4% | Sandy loam | 2.46 | 0.967 | 0.411 | 1.762 |
| ST3-DA | 1.55 | 15.6% | 8.4% | Loam | 1.86 | 0.711 | 0.437 | 1.856 |
| Vineyard: rows | 1.34 | 16.7% | 9.3% | Loam | 1.93 | 0.551 | 0.441 | 2.188 |
| Vineyard: interrow areas | 1.58 | 14.7% | 8.4% | Loam | 1.79 | 0.726 | 0.436 | 1.841 |
| Vineyard: corridor | 1.58 | 18.0% | 8.3% | Loam | 1.85 | 0.527 | 0.471 | 1.737 |
| Forest | 1.11 | 32.9% | 4.7% | Sandy loam | 4.64 | 1.980 | 0.354 | 1.497 |
| Bare soil trail | 1.63 | 16.3% | 6.5% | Sandy loam | 1.90 | 0.645 | 0.363 | 1.677 |
| Cereal field | 1.45 | 21.2% | 6.6% | Sandy loam | 1.96 | 0.401 | 0.458 | 1.720 |

BD: bulk density; SOM: soil organic matter; TN: total nitrogen; $P_{ava}$: available phosphorus; $K_{ava}$: available potassium; *: content from the fine fraction (ø < 2 mm); [1]: elaborated from [13,29].

### 2.4. Nutrient Enrichment Ratio and Statistical Analysis

The sediment/soil nutrient enrichment ratio ($ER_{nut}$) was calculated in the two sediment traps for each TIP:

$$ER_{nut} = \frac{Nut_{sed}}{\overline{Nut_{soil}}} \tag{1}$$

where $Nut_{sed}$ (mg of nutrients/g of sediment) is the concentration of nutrients ($TN_{sed}$ + $P_{ava-sed}$ + $K_{ava-sed}$) in the sediment, and $\overline{Nut_{soil}}$ (mg of nutrient/g of soil) is the average concentration of nutrients in the soil ($TN_{soil}$ + $P_{ava-soil}$ + $K_{ava-soil}$) of the drainage area of each sediment trap (ST2-DA and ST3-DA). Additionally, the ER of each nutrient and ST was calculated separately: $ER_{TN}$, $ER_{P-ava}$ and $ER_{K-ava}$ for ST2 and ST3.

In order to refine the data analysis, the loss of each nutrient at each ST and TIP was calculated considering two aspects: (I) the concentration of the nutrient in the sediment sample multiplied by the total amount of sediment yield; and (II) the extent of the drainage area of each ST, and the rates were expressed as $g_{nut}$/ha yr. All data were calculated at the annual and seasonal scales (low erosivity and ground cover, medium EI30 and GC, and high EI30 and GC). The statistical differences of the three nutrients, separately and jointly, between the two STs were analyzed over the test period by means of the analysis of variance (ANOVA; one-way) with the Shapiro–Wilk normality test at *p*-value < 0.05.

## 3. Results and Discussion

### 3.1. Soil Nutrient Content

The mean values of soil total nitrogen ($TN_{soil}$), available phosphorous ($P_{ava-soil}$) and potassium ($K_{ava-soil}$) in the drainage areas of ST2 and ST3 showed moderate changes between areas (Table 2). ST2-DA had higher content in $TN_{soil}$ (+36%) but lower content in $P_{ava-soil}$ (−6%) and potassium $K_{ava-soil}$ (−5%), compared with ST3-DA. The soil in ST2-DA had higher content in soil organic matter (+32%) but lower in very fine soil particles (clay and silt size; −7%) than the soil in ST3-DA. Regarding the different land uses and field compartments, the forest patches presented the highest content in SOM and $TN_{soil}$, while the vineyard had the highest values of $P_{ava-soil}$ and $K_{ava-soil}$. Within the vineyard, the interrow areas had a higher mean content of $TN_{soil}$ (+32%) but a lower mean content in SOM (−7%), $P_{ava-soil}$ (−1%) and $K_{ava-soil}$ (−16%) than the mean contents observed in the

rows. The small patches of arable land and bare soil presented intermediate values of SOM and nutrients.

The higher content in SOM and $TN_{soil}$ and lower content in $P_{ava-soil}$ and $K_{ava-soil}$ observed in ST2-DA compared with the values in ST3 was mainly due to the different characteristics and land uses of the drainage area of the two sediments traps. Small patches of forest occupy relevant parts within the DA of ST2 (22%), whereas the ST3-DA only had an average percentage of forest of 2%. The values of the SOM observed in the different land uses and field compartments agreed with the results of Jones et al. [31] and Ferreira et al. [32], who reported that Mediterranean soils generally have low SOM content (<2%) as a result of intensive use of the soils. In addition, higher SOM content was observed in the forest patches compared with the values in the vineyard, which has been reported in other studies in peninsular Spain [33], where the highest values of soil organic content were located under forest and scrubs and the lowest under agricultural soils (especially woody crops). Without fertilization, agricultural soils typically have lower organic matter and nutrient contents than natural soils. Within the vineyard, the corridor had similar content in SOM and N-P-K at the rows and interrow areas, despite the presence of spontaneous vegetation and the lack of ploughing operations, which confirms that the establishment of herbaceous vegetation and the cessation of ploughing may not be enough to trigger a recovery in soil organic matter content in cultivated areas, and changes in SOM are not readily reversed after ploughing ceases [34]. Regarding soil nutrient contents, the higher $P_{ava-soil}$ and $K_{ava-soil}$ observed in the vineyard could be partially explained by the inputs through mineral fertilization supplied just before planting the vines, but other processes such as topsoil mobility at short distances (e.g., washout erosion) and biological activity may have had a certain effect on the observed spatial heterogeneities. Despite the fact that phosphorous bioavailability depends on soil pH [35], previous edaphic studies showed that the soils in the landscape where our study area is located have a high content of calcium carbonate and the pH is alkaline [36]; therefore, the changes in $P_{ava-soil}$ among the different land covers and uses are not linked to inherent edaphic properties but to specific land management practices such as fertilization. However, a different behavior was observed in the content of TN due to the nitrogen being easily lost by leaching because of the nitrogen mobility in the soil profile [37].

### 3.2. Sediment Nutrient Content over the Study Period

During the study period, 139 rainfall events took place (111 events per year), accumulating 690 mm of rainfall depth (552 mm per year), but most events—58%—recorded fewer than 1 mm of precipitation per event, and can be considered as irrelevant in terms of soil erosion and nutrient loss (Table 3). Only 20% of the events (23 per year) had more than 8 mm of precipitation per event, and 14% of the events (16 per year) recorded more than 12 mm. Despite the study period being slightly drier than the average year, the number and characteristics of the erosive events were similar to the average conditions in the region (e.g., [38,39]). Therefore, the observed values of soil nutrient loss can be considered as representative of the prevalent conditions in the vineyard. From a total number of 26 field surveys or TIPs, we collected sediment in 21 cases, but the amount of sediment was very low in 7 TIPs, and thus, we measured the sediment nutrient content in 14 TIPs, adding up to 10 samples in ST2 and 14 samples in ST3. In a previous study on this site, Ben-Salem et al. [29] estimated the rainfall depth (12 mm) and erosivity (5.2 MJ mm/ha h event) thresholds for runoff (Q) initiation.

**Table 3.** Accumulated rainfall depth (ΣR) and erosivity (ΣEI30), level of ground cover (GC) and content of total nitrogen ($TN_{sed}$), available phosphorous ($P_{ava-sed}$) and available potassium ($K_{ava-sed}$) in the sediment collected in the two sediment traps (ST2 and ST3) at each time-integrated period (TIP) during the 15-month test period.

| TIP | | ΣR [†] | ΣEI [†] | GC | ST | SY [†] | $TN_{sed}$ | | $P_{ava-sed}$ | | $K_{ava-sed}$ | |
|---|---|---|---|---|---|---|---|---|---|---|---|---|
| Date | # | mm | MJ mm/ha h TIP | Level | # | g/TIP | $mg/g_{sed}$ | g/ha TIP | $mg/g_{sed}$ | g/ha TIP | $mg/g_{sed}$ | g/ha TIP |
| 07/02/2017 | 1 | 24.3 | 10.3 | Medium | ST2 | ND | ND | ND | ND | ND | ND | ND |
| | | | | | ST3 | 16.1 | 4.05 | 0.10 | 0.905 | 0.023 | 7.422 | 0.192 |
| 16/02/2017 | 2 | 30.1 | 30.4 | High | ST2 | ND | ND | ND | ND | ND | ND | ND |
| | | | | | ST3 | 39.8 | 2.11 | 0.14 | 0.921 | 0.059 | 3.343 | 0.214 |
| 08/03/2017 | 3 | 34.1 | 43.1 | High | ST2 | ND | ND | ND | ND | ND | ND | ND |
| | | | | | ST3 | 22.6 | 2.60 | 0.09 | 0.655 | 0.024 | 2.622 | 0.095 |
| 28/03/2017 | 4 | 70.3 | 68.1 | Very high | ST2 * | 30.9 | 3.14 | 0.29 | 1.249 | 0.117 | 3.944 | 0.370 |
| | | | | | ST3 * | 124.6 | 1.50 | 0.30 | 0.502 | 0.101 | 2.315 | 0.464 |
| 17/04/2017 | 5 | 11.7 | 5.1 | Very high | ST2 | 0 | ND | ND | ND | ND | ND | ND |
| | | | | | ST3 | 0 | ND | ND | ND | ND | ND | ND |
| 04/05/2017 | 6 | 23.3 | 29.4 | Very high | ST2 | 3.4 | IS | IS | IS | IS | IS | IS |
| | | | | | ST3 | 0 | ND | ND | ND | ND | ND | ND |
| 17/05/2017 | 7 | 31.9 | 68.8 | Very high | ST2 | 0 | ND | ND | ND | ND | ND | ND |
| | | | | | ST3 | 2.9 | IS | IS | IS | IS | IS | IS |
| 06/06/2017 | 8 | 39.9 | 27.9 | Low | ST2 | 2.3 | IS | IS | IS | IS | IS | IS |
| | | | | | ST3 | 7.1 | IS | IS | IS | IS | IS | IS |
| 16/06/2017 | 9 | 2.8 | 0.8 | Low | ST2 | 0 | ND | ND | ND | ND | ND | ND |
| | | | | | ST3 | 0 | ND | ND | ND | ND | ND | ND |
| 27/06/2017 | 10 | 19.4 | 47.6 | Low | ST2 | ND | ND | ND | ND | ND | ND | ND |
| | | | | | ST3 | 38.9 | IS | IS | IS | IS | IS | IS |
| 10/07/2017 | 11 | 10.8 | 8.0 | Low | ST2 | 1.7 | IS | IS | IS | IS | IS | IS |
| | | | | | ST3 | 1.4 | IS | IS | IS | IS | IS | IS |
| 29/08/2017 | 12 | 7.6 | 2.4 | Very low | ST2 | 0 | ND | ND | ND | ND | ND | ND |
| | | | | | ST3 | 0 | ND | ND | ND | ND | ND | ND |
| 19/09/2017 | 13 | 16.7 | 9.0 | Very low | ST2 | 27.8 | 4.38 | 0.37 | 0.916 | 0.077 | 2.804 | 0.237 |
| | | | | | ST3 | 15.2 | 5.06 | 0.12 | 0.755 | 0.018 | 2.847 | 0.070 |
| 26/09/2017 | 14 | 18.1 | 50.7 | Very low | ST2 ** | 32,825.8 | 0.45 | 44.95 | 0.409 | 40.847 | 1.161 | 115.929 |
| | | | | | ST3 | 1,821.6 | 2.29 | 6.71 | 0.717 | 2.101 | 2.188 | 6.414 |
| 17/10/2017 | 15 | 1.3 | 0.1 | Low | ST2 | 0 | ND | ND | ND | ND | ND | ND |
| | | | | | ST3 | 0 | ND | ND | ND | ND | ND | ND |
| 25/10/2017 | 16 | 42.9 | 107.9 | Low | ST2 ** | 41,260.2 | 0.63 | 79.11 | 0.362 | 45.479 | 1.057 | 132.746 |
| | | | | | ST3 ** | 2778.4 | 1.89 | 8.45 | 0.694 | 3.101 | 1.672 | 7.478 |
| 17/11/2017 | 17 | 8.0 | 15.3 | Medium | ST2 | 281.4 | 3.07 | 2.63 | 0.829 | 0.710 | 2.765 | 2.368 |
| | | | | | ST3 | 911.6 | 2.70 | 3.96 | 0.997 | 1.462 | 2.949 | 4.327 |
| 20/12/2017 | 18 | 13.8 | 2.9 | Medium | ST2 | 2.2 | IS | IS | IS | IS | IS | IS |
| | | | | | ST3 | 0.4 | IS | IS | IS | IS | IS | IS |
| 18/01/2018 | 19 | 28.6 | 12.1 | Medium | ST2 | 60.5 | 4.27 | 0.79 | 1.433 | 0.264 | 3.772 | 0.695 |
| | | | | | ST3 | 38.1 | 3.41 | 0.21 | 1.029 | 0.063 | 3.256 | 0.199 |
| 12/02/2018 | 20 | 42.4 | 11.5 | Medium | ST2 | 0.8 | IS | IS | IS | IS | IS | IS |
| | | | | | ST3 * | 44.6 | IS | IS | IS | IS | IS | IS |
| 19/02/2018 | 21 | 9.5 | 5.0 | High | ST2 | 0 | ND | ND | ND | ND | ND | ND |
| | | | | | ST3 | 0 | ND | ND | ND | ND | ND | ND |
| 07/03/2018 | 22 | 45.4 | 18.1 | High | ST2 * | 13.3 | 3.28 | 0.13 | 0.973 | 0.039 | 3.369 | 0.137 |
| | | | | | ST3 * | 18.8 | 3.01 | 0.09 | 0.914 | 0.028 | 2.681 | 0.081 |
| 19/03/2018 | 23 | 23.3 | 16.8 | Very high | ST2 | 84.5 | 4.29 | 1.10 | 1.154 | 0.297 | 4.517 | 1.161 |

**Table 3.** *Cont.*

| TIP | | ΣR [†] | ΣEI [†] | GC | ST | SY [†] | TN$_{sed}$ | | P$_{ava-sed}$ | | K$_{ava-sed}$ | |
|---|---|---|---|---|---|---|---|---|---|---|---|---|
| Date | # | mm | MJ mm/ha h TIP | Level | # | g/TIP | mg/g$_{sed}$ | g/ha TIP | mg/g$_{sed}$ | g/ha TIP | mg/g$_{sed}$ | g/ha TIP |
| | | | | | ST3 | 40.7 | 4.43 | 0.29 | 0.935 | 0.061 | 3.495 | 0.229 |
| 05/04/2018 | 24 | 17.7 | 5.6 | Very high | ST2 | 4.1 | IS | IS | IS | IS | IS | IS |
| | | | | | ST3 * | 42.0 | 3.17 | 0.21 | 1.080 | 0.073 | 4.543 | 0.307 |
| 18/04/2018 | 25 | 93.8 | 101.1 | Very high | ST2 * | 59.2 | 2.90 | 0.52 | 0.960 | 0.173 | 4.246 | 0.765 |
| | | | | | ST3 * | 30.2 | 3.42 | 0.17 | 0.905 | 0.044 | 3.501 | 0.170 |
| 30/04/2018 | 26 | 22.4 | 37.6 | Very high | ST2 * | 94.3 | 3.81 | 1.09 | 1.045 | 0.300 | 3.239 | 0.929 |
| | | | | | ST3 * | 572.7 | 5.82 | 5.36 | 0.928 | 0.855 | 3.050 | 2.811 |

[†]: data from [13,29]; ND: no data; IS: insufficient amount of sediment to perform the chemical analysis; *: ST completely full of runoff; **: ST completely full of sediments.

　　　The concentration of nutrients in the runoff water (see values of Q in Table 5 in [29]) differed between the two STs, with mean and median (between parenthesis) values of 479.8 (12.6), 316.2 (3.4) and 913.7 (12.0) mg of TN, P$_{ava}$ and K$_{ava}$/L TIP in the ST2 and 116.5 (6.3), 32.6 (1.8) and 99.8 (8.1) mg of TN, P$_{ava}$ and K$_{ava}$/L TIP in the ST3. The mean (median values between parentheses) concentration of nutrients in the sediment samples was of 3.02 (3.21) and 3.25 (3.09) mg of TN/g$_{sed}$ TIP, 0.93 (0.97) and 0.85 (0.91) mg of P$_{ava}$/g$_{sed}$ TIP and of 3.09 (3.30) and 3.28 (3.00) mg of K$_{ava}$/g$_{sed}$ TIP in the ST2 and ST3, respectively (Table 3). Therefore, a similar concentration of nutrients was found in the ST of the area with lower –ST2– and higher –ST3– GC, with mean and median (between parentheses) concentration of nutrients of 7.04 (8.10) and 7.38 (7.17) mg of N-P-K/g$_{sed}$ TIP in the ST2 and ST3. When the observed rates of nutrient loss were expressed accounting for the drainage area of each ST, the mean and median (between parentheses) rates were of 13.10 (0.94) and 1.87 (0.21) g TN/ha TIP, 8.83 (0.28) and 0.57 (0.06) g of P$_{ava}$/ha TIP and of 25.53 (0.85) and 1.65 (0.22) g of K$_{ava}$/ha TIP in the ST2-DA and ST3-DA, respectively. In total, the loss of nutrients was of 379.7 and 45.8 g of N-P-K/ha yr in the ST2-DA and ST3-DA. Gómez et al. [40] also found no significant differences in the nutrient loss in runoff (concentration of NO$_3$, P$_{soluble}$, and K$_{soluble}$) in Portuguese, Spanish and French olive groves managed with cover crops and under conventional tillage (weed removal), and significant differences only appeared when the total loss of soil was considered (see Table 5 of that study). Moreover, in a Mediterranean olive grove, López-Vicente et al. [14] found that the average eroded organic carbon (concentration in the sediment) was higher in the plots with low ground cover ($\overline{x}$ = 222 kg C$_{org}$/ha yr) compared with the plots with cover crop ($\overline{x}$ = 148 kg C$_{org}$/ha yr) plots, but differences were not significant among treatments. Overall, it seems that differences do not appear in the concentration of the chemical components in the sediment.

　　　A different picture appeared when all amounts of sediment yield were considered. The total loss of nutrients clearly differed between the two STs, with values of 43.04 and 16.29 g of TN, 29.02 and 4.98 g of P$_{ava}$ and of 83.90 and 14.32 g of K$_{ava}$ in the ST2-DA and ST3-DA. The net loss of nutrients was of 196.82 and 33.63 g of N-P-K in the ST2-DA and ST3-DA. These results clearly indicate the benefits of increasing the ground cover to preserve soil nutrients and how agronomic and environmental sustainability can be achieved by means of the functions associated with the mixed cover (temporary seeded cover crop with spontaneous vegetation). In vineyards located in eastern Croatia, Telak et al. [41] found that the sediment concentration, soil loss, carbon loss and P$_2$O$_5$ loss were significantly higher in the tilled plot than in the grass-covered plot. These authors highlighted that the tilled plots increased soil disaggregation and the availability of sediments for transport, enhancing C loss (4.5 times higher) and P$_2$O$_5$ loss (6.4 times higher) compared with the values in the grass-covered plots. In vineyards located in northern Italy, Biddoccu et al. [42] also found a clear reduction in the loss of nitrogen (NH$_4$-N and NO$_3$-N) and potassium (K) and comparable values of phosphorous (PO$_4$$^{3-}$) in the plots managed with grass cover

compared with the plots managed under conventional tillage and reduced tillage. Therefore, our results are consistent with those obtained in other vineyards under Mediterranean conditions. However, taking into account the differences in slope steepness between ST2-DA and ST3-DA, further research should be conducted in vineyards under homogeneous slope gradient conditions where all changes in SY and nutrient losses could be fully linked to changes in the percentage of GC.

Over the study period, the sediment nutrient concentration did not vary too much between the three periods of ground cover (GC), with mean values of 4.06, 8.07 and 8.42 mg of N-P-K/$g_{sed}$ TIP in ST2 during GC-low, GC-medium and GC-high, respectively, and of 6.04, 8.91 and 7.31 mg of N-P-K/$g_{sed}$ TIP in ST3 during GC-low, GC-medium and GC-high. However, clear changes of soil nutrient loss appeared during the three GC periods when the nutrient losses were expressed considering the total amounts of sediment yield and the drainage area of the two sediment traps (Figure 2a). Using this metric, the loss of nutrients was of 153.25, 3.73 and 1.49 g of N-P-K/ha TIP in ST2 during GC-low (from June to October), GC-medium (from November to mid-February) and GC-high (from mid-February to May), respectively. In ST3, the loss of nutrients was of 11.49, 3.51 and 1.53 g of N-P-K/ha TIP during GC-low, GC-medium and GC-high. On a monthly basis, the rates of nutrient loss were of 91.9, 2.1 and 2.1 g of N-P-K/ha month during GC-low, GC-medium and GC-high in the ST2-DA and of 6.9, 3.0 and 3.5 g of N-P-K/ha month during GC-low, GC-medium and GC-high in the ST3-DA (Figure 2b). Therefore, the average monthly loss of nutrients in the vineyard with lower ground cover (ST2-DA; 13.18 g of N-P-K/ha month) was 8.3 times higher than the rate in the vineyard with higher ground cover (ST3-DA; 1.59 g of N-P-K/ha yr). In vineyards located in northern Portugal (wet Mediterranean climate with a strong influence of the Atlantic Ocean), Ferreira et al. [4] monitored runoff and associated sediment and nutrient exports (total phosphorous—TP, total nitrogen—TN and nitrates—$NO_3$) in six runoff plots over two hydrological years, observing that about 60% of runoff and >85% of sediments and nutrients exported by runoff were recorded during winter, when the rainiest months are recorded. In our study area, there is a positive correlation between the evolution in the percentage of ground cover (GC) and the values of rainfall erosivity (EI30) over the course of the year. This fact is characteristic of the semiarid Mediterranean climate in the eastern part of Spain, avoiding the occurrence of intense events of soil loss during the rainy months of spring, but at the beginning of autumn—when heavy rainfall can occur—the GC is still limited and soil and nutrient losses can be high. Therefore, different seasonal combinations of GC and EI30 in other wine-growing regions around the world could lead to obtain distinct levels of benefits due to seeded cover crops. In Croatia, where rainfall is well-distributed over the year and differences in rainfall depth between the wet and dry seasons are not significant, Telak et al. [41] found that sediment loss and concentration were lower in the grass-covered plots than in the tilled plots in both the dry and wet seasons. In northern Italy, Biddoccu et al. [42] found that the highest differences of nutrient loss between the grass-covered and conventional tillage plots took place in winter and spring, with intermediate values of precipitation, but these differences became much less pronounced in autumn and summer, when the wettest and driest conditions happened.

### 3.3. Nutrient Enrichment Ratio

Most enrichment ratios were above 1.0, indicating a clear accumulation of soil nutrients in the sediments (Figure 3a). The mean $ER_{TN}$ was of 3.12 and 4.57 in ST2 and ST3, respectively, whereas the mean $ER_{Pava}$ was of 2.27 and 1.95 in ST2 and ST3, and the mean $ER_{Kava}$ was of 1.75 and 1.77 inST2 and ST3. These results show that there was not a clear difference between the two sediment traps because the average $ER_{N-P-K}$ was of 2.38 and 2.76 in ST2 and ST3. These similarities mirrored the patterns of sediment nutrient concentrations, with comparable values between the two areas, with high and very high ground cover. Our results agree with those obtained by Farsang et al. [16] in Hungarian vineyards, where the ERs of organic matter (OM) ($ER_{OM}$ = 2.1–2.2) and $P_2O_5$ ($ER_P$ = 1.8–2.1) were quite similar. In three Spanish vineyards, Ruiz-Colmenero et al. [17] also obtained

a mean ER higher than 1.0 for the content of organic matter ($ER_{OM}$ = 1.4–2.0), nitrogen ($ER_N$ = 1.0–2.3) and phosphorus ($ER_P$ = 1.1–2.2).

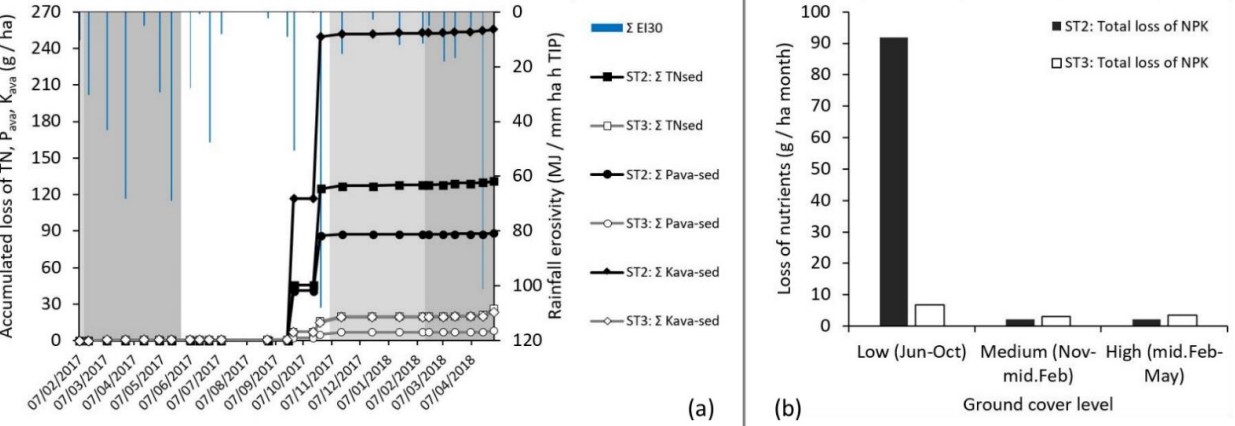

**Figure 2.** Accumulated loss of total nitrogen, available phosphorous and available potassium over the study period in the two sediment traps (STs), including the cumulative rainfall erosivity per time-integrated period (TIP)—the white, light gray and dark gray backgrounds correspond to the periods of low, medium and high ground cover, respectively (**a**). Total loss of nutrients (N-P-K) during the three periods of ground cover (GC) in the two STs (**b**).

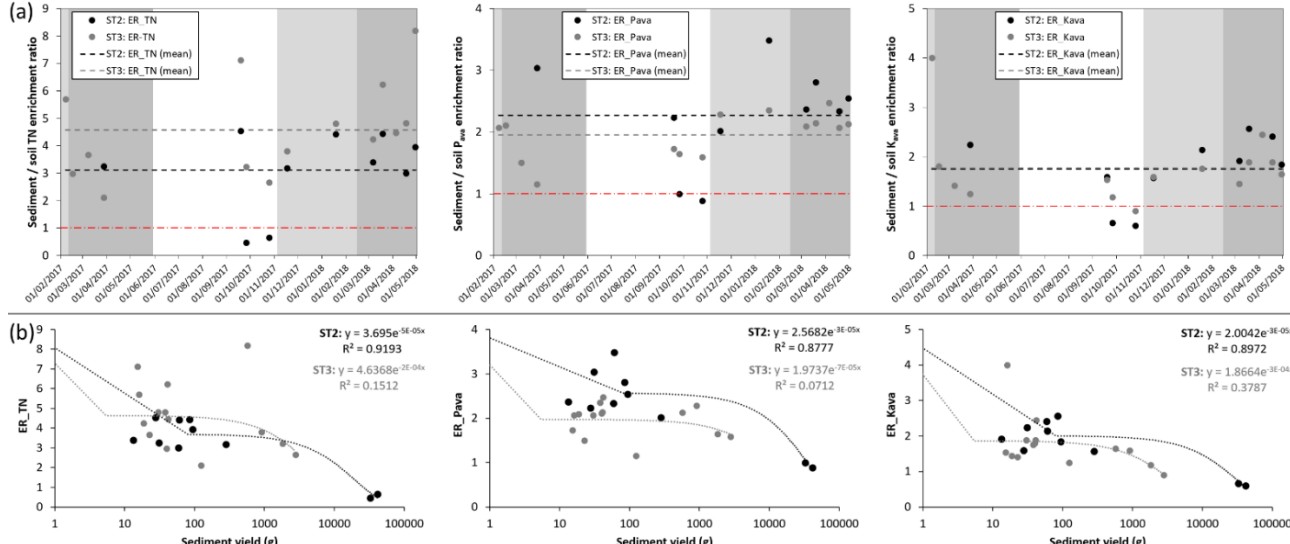

**Figure 3.** Evolution of the sediment/soil nutrient enrichment ratio (ER) of nitrogen, phosphorous and potassium (**a**). Correlation between the ER of the three nutrients and the sediment yield at each TIP (**b**). The white, light grey and dark grey backgrounds correspond to the periods of low, medium and high ground cover, respectively.

Over the course of the studied period the ER varied, the lowest ratios ($ER_{N-P-K}$ = 1.40 and 2.40 in ST2 and ST3) appearing during the months with the lowest ground cover (GC), when the highest sediment yields were obtained. Conversely, during the months with medium and high GC, which coincided with the lower sediment yields, the sediment/soil enrichment ratios were higher: $ER_{N-P-K}$ = 2.80 and 3.15 in ST2 and ST3 when GC-medium and $ER_{N-P-K}$ = 2.80 and 2.75 in ST2 and ST3 when GC-high. Therefore, there is an inverse relationship between the magnitude of soil erosion and the accumulation of nutrients in the sediments (Figure 3b). This relationship was stronger in the area with lower GC and higher SY (ST2) and much less marked in ST3 (higher soil protection), indicating that the effect of rainfall erosivity on the sediment/soil nutrient enrichment ratio is higher when GC is

low, and this influence decreases as GC increases. In a Chinese farmland, Jin et al. [43] also found that higher rainfall intensity and lower cover produced higher sediment, and consequently higher nutrient loss, but resulted in a lower soil organic carbon enrichment ratio ($ER_{SOC}$) in the sediment. Therefore, the amount of runoff sediment rather than the ER in the sediment was the determining factor for the amount of nutrients lost. This fact may be explained due to the particle size selectivity of the soil erosion process. In calcareous colluvial soils, Martínez-Mena et al. [44] found that rainfall intensity affected the predominant erosion process, appearing in detachment-limited conditions during high-intensity events (very high connectivity and long travel distance) and transport-limited conditions during medium-intensity events (moderate connectivity and shorter travel distance). Therefore, a greater amount of coarse materials, which are poorer in nutrient content, may be transported during the high-intensity events of the GC-low period, while a higher amount of fine materials, which are richer in nutrients, may be delivered during the events of the GC-medium and GC-high periods.

*3.4. Further Research*

In order to obtain sounder results of soil nutrient loss and sediment/soil nutrient enrichment ratios ($ER_{nutrient}$) in vineyards, we plan to extend the analysis to other vineyards and/or woody crops—olive groves and fruit tree orchards—with different ground covers such as straw mulch, spontaneous vegetation and pruning residues and different types of seeded cover crops (permanent/temporary, homogeneous/mixed and mowing/ ploughing). The literature about $ER_{nutrient}$ in woody crops is limited, and thus, this study provides new data that should be supplemented with new findings from other case studies located in areas with distinct physiographic conditions. The survey of vineyards' soil erosion has been receiving more attention since the beginning of the 21st century based on different methodologies: rainfall simulations [45], runoff and sediment plots [46], topographical measurements [47] and modelling [48]. Within that research perspective, the study of the connectivity of water and sediments in vineyards has been a key topic to understand the mechanism of sediment transport [29,49]. Measurements of the solutes have been rare in vineyards and our research is pioneering. More than a survey of solutes release, this paper aims to initiate a new research objective to better understand the soil losses within the sustainability challenge in agriculture. This paper contributes to furthering the knowledge of soil erosion and sustainability in vineyards in three ways: (I) to report the soil losses due to solutes losses; (II) to contribute to better understand the soil erosion process and mechanism, highlight the opportunity that solutes bring to be used as tracers of soil erosion and understand the connectivity of water and sediments; and (III) to shed light about management that can contribute to a better soil management to preserve nutrients and reduce solute release.

An implication of our findings is that solutes are easily transported by the overland flow in vineyards. This must encourage the application of management that will reduce solute release to avoid damages in ecosystems that are sensitive to an increase in nutrients such as the wetlands and rivers. Moreover, our research also advises about the risk of delivering potentially toxic elements, such as Manaljav et al. [50] found in the sloping vineyards and Pham et al. [51] in Tokaj Nagy Hill, both sites in Hungary. Runoff water transports nutritive elements from soil, but they can also mobilize toxic elements [52]. The probability of transport of a certain element in soil is greatly influenced by its chemical speciation and increases when the element is in an easy mobilized form, being that dissolved nutrient forms easily mobilized with any runoff water [53]. The loss of soil is a key factor in the sustainability of agriculture land. Without a healthy soil, the crop production is dependent on chemical fertilizers. Today, after the COVID-19 pandemic and within the war in Ukraine, humankind has found that the need to reduce the use of chemical fertilizers and to protect the health of the soils is also a strategy to achieve safe food production [54]. We propose to extend the use of cover crops in vineyards to reduce soil losses. Additionally,

the use of catch crops and cover crops is sustainable as they grow in the same field and their impact also improves soil properties [55] and closes the nitrogen cycle [56].

Based on our results, the main benefit of the seeded cover crop was to reduce the total amount of soil loss rather than reduce the concentration of nutrients in the sediment that remained with similar values in the areas with lower and higher percentages of ground cover. However, the observed temporal dynamic of soil loss, total nutrient loss and $ER_{N-P-K}$ with clear differences between the periods with low, medium and high ground cover and / or rainfall erosivity suggests the necessity of refining the experimental design in order to identify the areas within the field that make a greater contribution to the net loss of soil and nutrients. Other aspects that should be included in further research are: (I) the effect of (selective) nutrient infiltration into the subsoil or nearby water flows [57]; (II) the potential concentration of some nutrient/s in specific ranges of soil and sediment particle size [58]; and (III) the effect of the continued process of soil nutrient impoverishment on the availability of soil nutrients to be delivered [59].

## 4. Conclusions

This paper evaluates the effectiveness of a mixed ground cover (GC) made up of a temporary seeded cover crop and spontaneous vegetation to reduce the loss of soil nutrients (N-P-K) in a rainfed vineyard. Sediment nutrient concentrations and sediment/soil enrichment ratios were measured and calculated over 15 months, showing important changes over the course of the study period. The loss of nutrients was 8.3 times higher in the soil with a lower percentage of ground cover. The benefits of greater GC promote the environmental and agronomic sustainability by the functions of the cover crop and weeds that protect the soils from soil losses and enhance the nutrient enrichment in the soil. The use of GC resulted in a reduction in nutrients lost that contribute to a more sustainable agriculture with more nutrients available for the crops, healthy soils and a reduction in the investment of farmers in fertilizers. This is very relevant in a postpandemic world under the threat of the war in Ukraine and the lack/prolonged uncertainty of access to fertilizers and the need for a local production of food.

**Supplementary Materials:** The following supporting information can be downloaded at: https://www.mdpi.com/article/10.3390/land11071033/s1, Figure S1: Map of the slope gradient in the study area.

**Author Contributions:** Conceptualization and formal analysis, M.L.-V.; field surveys and laboratory work, M.L.-V., S.Á. and E.C.-S.; writing—original draft preparation, review and editing, M.L.-V., S.Á. and A.C. All authors have read and agreed to the published version of the manuscript.

**Funding:** The Spanish Ministry of Economy and Competitiveness funded this research through the project "Environmental and economic impact of soil loss (soil erosion footprint) in agro-ecosystems of the Ebro river basin: numerical modelling and scenario analysis (EroCostModel) (CGL2014-54877-JIN)".

**Institutional Review Board Statement:** Not applicable.

**Informed Consent Statement:** Not applicable.

**Data Availability Statement:** The data that support the findings of this study are available from the corresponding author upon reasonable request.

**Acknowledgments:** We thank Gonzalo Alcalde Fábregas (Fábregas winery, DOP Somontano) for permitting the use of the vineyards where this research was conducted.

**Conflicts of Interest:** The authors declare no conflict of interest.

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
