# Peer review of "Sustainability by Function (SbF): A Case Study in a Rainfed Vineyard to Reduce the Loss of Soil Nutrients"

_land, doi:10.3390/land11071033_

Round 1
Reviewer 1 Report
Dear authors,
Please check all the comments by the reviewer as attached (Reviewer's comments), and reply each of them.
Reviewer

Author Response
Reply to Reviewer #1
Comments and Suggestions for Authors
Dear authors,
Please check all the comments by the reviewer as attached (Reviewer's comments), and reply each of them.
Reply to R1: Thank you for your time and work; your comments are valuable to improve the manuscript.
The present authors experimentally investigated the process of soil nutrient decline in vineyard soils to add new data for establishing sustainable tillage practices. The aim and experiments in the field are to be appreciated. However, discussion may be partly skewed without using all the data obtained. As the title indicated, the term “sustainability by function” should be denoted in discussion (or conclusion).
Reply to R1: We have improved the manuscript by means of addressing all your comments, including a greater use of concept “sustainability by function” in the Discussion. However, we already included the concept “sustainability by function” in the Conclusions section of the original manuscript.
Introduction. Page 2, line 92: where two sediment traps where were installed
Reply to R1: Thank you. We have corrected the sentence.
Introduction. Page 2, line 79: SdD refers to how… The term SbD refers to
Reply to R1: Thank you. We have corrected the sentence.
Introduction. Page 3, line 144: Mean slope gradient was quite different between ST2 (17.0 %) and ST3 (9.2 %). A topographical diagram may be useful for better understanding which direction the slope was oriented at ST2 and ST3.
Reply to R1: We have expanded the content of the manuscript regarding the slope gradient of the drainage area of the two ST, as follows: “Despite these differences between the mean slope gradients, the mean slope near the STs was very similar in the two ST-DA (ca. 9.5%). Besides, the mean slope in the rows (where bare soil conditions were predominant in the two ST-DA) were also comparable in ST2-DA and ST3-DA. The highest values of slope gradient in ST2-DA only appeared in the small forest patch (mean slope = 18.4%) and in the inter-row areas near the forest (Supplementary file 1). Therefore, we consider that the impact of the different slope gradients on the total sediment yield and nutrient loss is less relevant than the differences in the percentage of ground cover.” We have added a new map to show the slope gradient. Besides, we have added three new columns in Table 1 in order to reflect the values of slope gradient near the STs, and in the rows and inter-row areas. Regarding the overland flow pathways, the original figure 1 clearly shows the runoff spatial pattern.
Materials and Methods. Page 5, Table 1: The footnote erroneously denoted vineyard as VN, but actually VY in the table?
Reply to R1: Thank you. We have corrected the sentence.
Results and discussion. Page 7, lines 267-270: Was the different net loss between drainage areas at ST2 and ST3 only due to the ground cover? How about the difference in slopes between them in which it was 17 % at ST2.
Reply to R1: We agree with R1 than slope gradient is an important factor that requires a special attention, and thus, we have added the following sentence: “However, and taking into account the differences in slope steepness between ST2-DA and ST3-DA, further research should be done in vineyards under homogeneous slope gradient conditions where all changes in SY and nutrient losses could be fully linked to changes in the percentage of GC.”.
Results and discussion. Page 9, Figure 2 (a): Symbols in the figure are difficult to be distinguished. Closed and open shapes should be used in monotone.
Reply to R1: We have improved the layout of the figure, in order to clearly distinguish the values that correspond to ST2 and ST3.
Results and discussion. Page 9, lines 334-335: Was there any experimental evidence to support this statement?
Reply to R1: This statement is totally based on our field observations and data. Besides, Figure 3b shows the mathematical relationship between the values of sediment yield and the enrichment ratios of the three nutrients. To support our results, we have added the following comment and reference: “In a Chinese farmland, Jin et al. [43] also found that higher rainfall intensity and lower cover produced higher sediment and consequently higher nutrient loss, but resulted in a lower soil organic carbon enrichment ratio (ERSOC) in the sedi-ment. Therefore, the amount of runoff sediment rather than the ER in the sediment was the determinant factor for the amount of nutrients lost.”. 43. Jin, K.; Cornelis, W.M.; Gabriels, D.; Baert, M.; Wu, H.J.; Schiettecatte, W.; Cai, D.X.; De Neve, S.; Jin, J.Y.; Hartmann, R.; Hofman, G. Residue cover and rainfall intensity effects on runoff soil organic carbon losses. Catena 2009, 78 (1), 81–86.
Results and discussion. Page 9, lines 339-341: The authors described in the experimental section (Page 4, lines 155-170) that the runoff and sediment samples were collected. However, they did not show any results, such as varying concentration of nutrients on the runoff (drain) waters. It must be much better to discuss all the results obtained in this study comprehensively.
Reply to R1: All results of sediment yield were already included in Table 3. Besides, the concentration and loss of each nutrient (N, P and K) in the sediment, during each field survey and considering the extension of each drainage area, was also presented in this table. The amount of runoff was published in a previous study and the reference was included in the manuscript. Therefore, we cannot agree with this comment because all data was presented with transparency.
Results and discussion. Page 10, Figure 3 (b): The figures were not clear especially in the lower sediment yield in which the regression line between sediment yield and enrichment ratio of nutrient was quite poor in ST3. How was the non-linear regression curve drawn for each nutrient in ST2?
Reply to R1: We have improved the layout of the charts and added clear information of which correlation corresponds to each ST.
Results and discussion. Page 11, lines 365-383: Speciation of soil nutrients is important to determine how they are drained out in the field, i.e., dissolved form or association with particulate matter in soil.
Reply to R1: Thank you for this comment. We have clarified this aspect in the revised version, as follows (underlined text): “(…) Runoff water transports nutritive elements from soil, but they can also mobilize toxic elements [52]. The probability of transport of a certain element in soil is greatly influenced by its chemical speciation and increases when the element is in an easy mobilized form, being dissolved nutrient forms easily mobilized with any runoff water [53]. The loss of soil is a key factor on the sustainability of agriculture land. Moreover, one of the main challenges in agriculture nowadays is to reduce the nutrient losses and mitigate drainage effects. Without a healthy soil, the crop production is dependent on the chemical fertilizers.”
- Barberis, E.; Celi, L.; Martin, M. Speciation and bioavailability of soil nutrients: effect on crop production and environment. Italian Journal of Agronomy 2009, 4(s1), 23–32.
- Baulch, H.M.; Elliott, J.A.; Cordeiro, M.R.C.; Flaten, D.N.; Lobb, D.A.; Wilson, H.F. Soil and water management: opportunities to mitigate nutrient losses to surface waters in the Northern Great Plains. Environmental Reviews 2019, 27, 447 – 477.
Reviewer 2 Report
The study reports a comparative analysis of seeded cover crop to minimize soil nutrient losses in rainfed vineyard was evaluated in a rainfed vineyard. It supply two sediment tanks with high ground cover 82% for one and 89% for the other, and samples were collected from 26 time-integrated periods over 15 months. The study also attempt to quantify the soil nutrient loss for different cover rate and give some practical suggestions for the agriculture management. The results show that loss of soil nutrients in lower ground cover was 8.3 times higher than in the high ground cover. The study supplied the scientific data for the benefit of ground cover to agriculture in the rainfed area. It is very practical study in the agriculture system. Some small revisions need to be clarified.
Major points:
1. There are two concepts need to be clarified. One is the quantity of sediment from land, the other is soil nutrients for sediment. The quantity of sediment in the area with high ground cover is lower which is easy to be understood by readers, while the soil nutrients was lower in the high ground cover which need to be logically explained. The difference is due to the different nutrient content themselves or some other reasonable biological process.
2. The site for ST2 and ST3 are different and the slope is also different. How can you balance the difference of soil nutrients resulting from the slope, not by the ground cover?
3. The soil nutrients loss were low that may be related to the high efficiency of rainfall interception of the high ground cover, then lead to high soil moisture, and high microbial activity. So if you have the data for soil moisture and microbial, that would be more reasonable.
Author Response
Reply to Reviewer #2
Comments and Suggestions for Authors
The study reports a comparative analysis of seeded cover crop to minimize soil nutrient losses in rainfed vineyard was evaluated in a rainfed vineyard. It supply two sediment tanks with high ground cover 82% for one and 89% for the other, and samples were collected from 26 time-integrated periods over 15 months. The study also attempt to quantify the soil nutrient loss for different cover rate and give some practical suggestions for the agriculture management. The results show that loss of soil nutrients in lower ground cover was 8.3 times higher than in the high ground cover. The study supplied the scientific data for the benefit of ground cover to agriculture in the rainfed area. It is very practical study in the agriculture system. Some small revisions need to be clarified.
Reply to R2: Thank you for your positive evaluation and valuable comments to improve the manuscript.
Major points:
- There are two concepts need to be clarified. One is the quantity of sediment from land, the other is soil nutrients for sediment. The quantity of sediment in the area with high ground cover is lower which is easy to be understood by readers, while the soil nutrients was lower in the high ground cover which need to be logically explained. The difference is due to the different nutrient content themselves or some other reasonable biological process.
Reply to R2: Only the content of total nitrogen and organic matter was higher in the soils of the drainage area of ST2 (lower ground cover), while the content of phosphorous and potassium was higher in ST3-DA. As we said in the original manuscript, the higher concentrations in TN and SOC in ST2-DA was associated with the presence of forest areas. The small patches of forest occupy relevant parts within the DA of ST2 (22%), whereas the ST3-DA only had an average percentage of forest of 2%.
- The site for ST2 and ST3 are different and the slope is also different. How can you balance the difference of soil nutrients resulting from the slope, not by the ground cover?
Reply to R2: We have expanded the content of the manuscript regarding the slope gradient of the drainage area of the two ST, as follows: “Despite these differences between the mean slope gradients, the mean slope near the STs was very similar in the two ST-DA (ca. 9.5%). Besides, the mean slope in the rows (where bare soil conditions were predominant in the two ST-DA) were also comparable in ST2-DA and ST3-DA. The highest values of slope gradient in ST2-DA only appeared in the small forest patch (mean slope = 18.4%) and in the inter-row areas near the forest (Supplementary file 1). Therefore, we consider that the impact of the different slope gradients on the total sediment yield and nutrient loss is less relevant than the differences in the percentage of ground cover.” We have added a new map to show the slope gradient. Besides, we have added three new columns in Table 1 in order to reflect the values of slope gradient near the STs, and in the rows and inter-row areas. Regarding the overland flow pathways, the original figure 1 clearly shows the runoff spatial pattern.
- The soil nutrients loss were low that may be related to the high efficiency of rainfall interception of the high ground cover, then lead to high soil moisture, and high microbial activity. So if you have the data for soil moisture and microbial, that would be more reasonable.
Reply to R2: We already mentioned two previous studies done in the same site that demonstrated the effectiveness of the ground cover to favour soil water infiltration and increase the soil water content [28], and to reduce the loss of soil and sediment turbidity [29] and of soil organic matter [13].
Reviewer 3 Report
Presented study aims the problem of nutrient decline due to erosion process describing very important topic of nutrient management and predicting the potential losses on nutrients depending on land use type and agronomic activities. In my opinion this research is novel and valuable, having an important impact on the state of knowledge about this urgent topic. I’ve also liked the idea of using SbD concept as a tool for predictions and is very interesting in the context of soil fertility modeling. I can fully recommend this work for publication.

Author Response
Reply to Reviewer #3
Comments and Suggestions for Authors
Presented study aims the problem of nutrient decline due to erosion process describing very important topic of nutrient management and predicting the potential losses on nutrients depending on land use type and agronomic activities. In my opinion this research is novel and valuable, having an important impact on the state of knowledge about this urgent topic. I’ve also liked the idea of using SbD concept as a tool for predictions and is very interesting in the context of soil fertility modeling. I can fully recommend this work for publication.
Reply to R3: Thank you for your positive evaluation. We appreciate your words.
L.64-65: Is it necessary to use "soil" multiple times?
Reply to R3: We have simplified the text.
- 231-232: What was the soil pH important for P bioavailability? Is it possible to indicate P and K after so many years from fertlization and plant production?
Reply to R3: We totally agree with this comment. P bioavailability depends on soil pH, as inherent soil properties and climate affect how crops respond to applied P fertilizer, and regulate processes that limit P availability. Phosphorous (P) is a plant nutrient and its availability appears to be affected directly by soil pH (Hinsinger, 2001). Unfortunately, we did not measure soil pH in our study. However, previous studies determined that soils in the vineyards of the DOP Somontano located near Barbastro, like the one of our study, have a high content of calcium carbonate and pH is alkaline (Casanova-Gascón et al., 2018). We have added this information and the new reference in the revised manuscript. In our study, the higher Pava and Kava in the vineyards can be due to several causes, fertilization, mobility (erosion or sediment deposition), soil biological processes, etc. In the manuscript we write that initial fertilization can be one of these causes. Apart from the other causes and the time elapsed since supply of P and K, the soil in the vineyard had an extra amount of P before planting and the initial content was higher than in the rest of uses (forest or bare soil trail). We have improved the text as follows: “Regarding soil nutrient contents, the higher in Pava-soil and Kava-soil observed in the vineyard could be partially explained by the inputs through mineral fertilization supplied just before planting the vines, but other processes like topsoil mobility at short distances (e.g. washout erosion) and biological activity may have had a certain effect on the observed spatial heterogeneities. Despite the fact that phosphorous bioavailability depends on soil pH [35], previous edaphic studies showed that the soils in the landscape where our study area is located have a high content of calcium carbonate and pH is alkaline [36]; therefore, the changes in Pava-soil among the different land covers and uses are not linked to inherent edaphic properties, but to specific land management practices like fertilization.”.
[35] Hinsinger, P. Bioavailability of soil inorganic P in the rhizosphere as affected by root-induced chemical changes: a review. Plant and Soil (2001) 237, 173–195. https://doi.org/10.1023/A:1013351617532
[36] Casanova-Gascón J, Martín-Ramos P, Martí-Dalmau C, Badía-Villas D. Nutrients Assimilation and Chlorophyll Contents for Different Grapevine Varieties in Calcareous Soils in the Somontano DO (Spain). Beverages (2018) 4 (4), article number 90; https://doi.org/10.3390/beverages4040090
Round 2
Reviewer 1 Report
Most comments by the reviewer were checked and revised properly.
However, they did not reply for the following two comments yet.
They should at least add appropriate explanation in the text.
Page 9, lines339-341: The authors described in the experimental section (Page 4, lines 155-170) that the runoff and sediment samples were collected. However, they did not show any results, such as varying concentration of nutrients on the runoff (drain) waters. It must be much better to discuss all the results obtained in this study comprehensively.
Page 10, Figure 3 (b): The figures were not clear especially in the lower sediment yield in which the regression line between sediment yield and enrichment ratio of nutrient was quite poor in ST3.
The authors should explain possible causes of such a low correlation between enrichment ratio of three nutrients and sediment yield at ST3.
By the way,
Page 2, line 78: The term SbD refers to
Reviewer
Author Response
Reply to Reviewer #1
Comments and Suggestions for Authors
Most comments by the reviewer were checked and revised properly. However, they did not reply for the following two comments yet. They should at least add appropriate explanation in the text.
Reply to R1: Thank you for your comments. We have replied to all of them, improving the quality and soundness of the text.
Page 9, lines 339-341: The authors described in the experimental section (Page 4, lines 155-170) that the runoff and sediment samples were collected. However, they did not show any results, such as varying concentration of nutrients on the runoff (drain) waters. It must be much better to discuss all the results obtained in this study comprehensively.
Reply to R1: Using the values of runoff water (already published in [29]), we have calculated the concentration of nutrients per liter and time-integrated period (in mg / L TIP), and improved the text as follows: “The concentration of nutrients in the runoff water (see values of Q in Table 5 in [29]) differed between the two STs, with mean and median (between parenthesis) values of 479.8 (12.6), 316.2 (3.4) and 913.7 (12.0) mg of TN, Pava and Kava / L TIP in the ST2, and 116.5 (6.3), 32.6 (1.8) and 99.8 (8.1) mg of TN, Pava and Kava / L TIP in the ST3.”
Page 10, Figure 3 (b): The figures were not clear especially in the lower sediment yield in which the regression line between sediment yield and enrichment ratio of nutrient was quite poor in ST3.
Reply to R1: We have improved the layout of the charts by using logarithmic scale in the X-axis of the relationships between sediment yield (SY) and the sediment/soil nutrient enrichment ratios (ER). Now, all values near zero can be seen very clearly.
The authors should explain possible causes of such a low correlation between enrichment ratio of three nutrients and sediment yield at ST3.
Reply to R1: We have improved the text of the manuscript as follows: “This relationship was stronger in the area with lower GC and higher SY (ST2), and much less marked in ST3 (higher soil protection), indicating that the effect of rainfall erosivity on the sediment/soil nutrient enrichment ratio is higher when GC is low, and this influence decreases as GC increases.”
By the way, Page 2, line 78: The term SbD refers to
Reply to R1: Thank you for identifying this writing mistake; we have corrected the text.